# Attention to the strengths of physical interactions: Transformer and graph-based event classification for particle physics experiments

Luc Builtjes[1], Sascha Caron[1,2*], Polina Moskvitina[1,2†], Clara Nellist[2,3‡], Roberto Ruiz de Austri[4°], Rob Verheyen[5§] and Zhongyi Zhang[1,2,6‖]

**1** High Energy Physics, Radboud University Nijmegen, Heyendaalseweg 135, 6525 AJ Nijmegen, the Netherlands
**2** Nikhef, Science Park 105, 1098 XG Amsterdam, the Netherlands
**3** Institute of Physics, University of Amsterdam, 1090 GL Amsterdam, The Netherlands
**4** Instituto de Física Corpuscular, IFIC-UV/CSIC, Valencia, Spain
**5** Department of Physics and Astronomy, University College London, London, WC1E 6BT, UK
**6** The Bethe Center for Theoretical Physics, Bonn University, 53115 Bonn, Germany

⋆ scaron@nikhef.nl , † p.moskvitina@nikhef.nl , ‡ c.nellist@nikhef.nl , ○ rruiz@ific.uv.es , § r.verheyen@ucl.ac.uk , ‖ zhongyi@th.physik.uni-bonn.de

## Abstract

A major task in particle physics is the measurement of rare signal processes. These measurements are highly dependent on the classification accuracy of these events in relation to the huge background of other Standard Model processes. Reducing the background by a few tens of percent with the same signal efficiency can already increase the sensitivity considerably. This work demonstrates the importance of incorporating physical information into deep learning-based event selection. The paper includes this information into different methods for classifying events, in particular Boosted Decision Trees, Transformer Architectures (Particle Transformer) and Graph Neural Networks (Particle Net). In addition to the physical information previously proposed for jet tagging, we add particle measures for energy-dependent particle-particle interaction strengths as predicted by the leading order interactions of the Standard Model (SM). We find that the integration of physical information into the attention matrix (transformers) or edges (graphs) notably improves background rejection by 10% to 40% over baseline models (a graph network), with about 10% of this improvement directly attributable to what we call the SM interaction matrix. In a simplified statistical analysis, we find that such architectures can improve the significance of signals by a significant factor compared to a graph network (our base model).

## Contents

---

## 1  Introduction

With the unprecedented amount of data provided by the upcoming runs of the Large Hadron Collider (LHC), one can start to measure rare processes with very small cross-sections. Examples are the recent observations of four top quarks originating from a single proton-proton collision event [1, 2]. At the heart of this endeavor is the difficult task of detecting and measuring rare signaling processes amidst the overwhelming background noise generated by the multitude of Standard Model processes. Accurate classification of these events is crucial, as even a small reduction in background noise on the order of a few tens of percent while maintaining the same signal detection efficiency can lead to a profound increase in sensitivity.

At the same time, the recent deep learning revolution has found a large variety of applications in high energy physics (see e.g. [3] for a review). One of these is the development of a large variety of architectures for the purpose of classification of particle physics data, including improved BDTs [4], convolutional networks [5], graph neural networks [6] and attention-based architectures [7, 8]. These methods are mainly used in the context of the classification of jet data. On the other hand, their application to event-level data has not yet been explored to the same degree, and BDTs are still the most commonly used method.

In this work, we perform a comparison of different event classification methods and highlight a crucial aspect: the inclusion of physical information and inductive biases in machine learning architectures. In addition to the already proposed data used for jet tagging, we introduce a new approach by incorporating particle measures that capture the subtleties of particle-particle interaction strengths as predicted by the Feynman rules of the Standard Model [1] With this in mind, we present a publicly available dataset to test classification methods with four top and top-top-Higgs events. We perform wide hyperparameter scans over all models and compute several performance metrics to evaluate their performance.

In Section 2, we describe the data generation and the data format. Section 3 briefly describes the Machine Learning (ML) models used in the comparison and their optimization. In Section 4, we explore how to inform the ML models about physics. We discuss our results in Section 5, followed by the conclusions in Section 6.

# 2 Data description

In this section, we describe the data generation and the data format used for this work.

## 2.1 Data generation

We simulated proton-proton collisions at a center-of-mass energy of 13 TeV. The relevant production processes for this work consist of the backgrounds $t\bar{t}$ + X, where $X = Z, W^+$ and $W^+W^-$ and signal processes including the four top production process and $t\bar{t}H$ production. The hard scattering process generation was performed at leading order, where up to two additional jets are added to the final state in the case of the background processes, and up to one for the signal. The cross-sections for all the processes and the corresponding total number of events generated are depicted in Tab. 1.

The hard scattering was generated with `MG5_aMC@NLO` version 2.7 [9] with the `NNPDF31_lo` parton distribution functions set [10], using the **5** flavor scheme. Parton showering was performed with `Pythia` version 8.239 [11], while the MLM merging scheme [12] was used to merge high-multiplicity hard scattering events with the parton shower. A fast detector simulation was performed with `Delphes` version 3.4.2 [13] using the ATLAS detector map. Finally, an $H_T = \sum_{jets} E_T > 400$ GeV restriction was imposed at the parton level during event generation. The purpose of this restriction is generation efficiency, since our signal region imposes $H_T > 500$ GeV on the reconstructed objects (see Section 2.2 for details.).

| Physics process | $\sigma$ (pb) | $N_{\text{tot}}$ | $\epsilon$ |
|---|---|---|---|
| $pp \to t\bar{t}t\bar{t}$ (+1 j) | 0.01 | 32463742 | 0.007 |
| $pp \to t\bar{t}h$ (+2 j) | 0.022 | 29783343 | 0.001 |
| $pp \to t\bar{t}W^{\pm}$ (+2 j) | 0.045 | 8954246 | 0.005 |
| $pp \to t\bar{t}W^+W^-$ (+2 j) | 0.0096 | 20160377 | 0.003 |
| $pp \to t\bar{t}Z$ (+2 j) | 0.034 | 10605846 | 0.011 |

Table 1: Signal and background processes with the corresponding LO (Leading Order) cross-section $\sigma$ in pb (second column), the total number of generated events $N_{\text{tot}}$ (third column) and the efficiency $\epsilon$ of the cuts applied (fourth column).

---

[1]Pairwise particle-particle interaction strengths and 4-vector correlations are called "pairwise kinematic features" in the following text, see chapter 4 for details.

### 2.2 Event and object selection

Collision events consist of objects such as jets, b-jets, leptons, and photons, each with their corresponding kinematic variables (see Section 2.3). Following the strategy in Ref. [14], an event is saved if at least one of the following conditions is met:

- At least one jet or a $b$-jet with transverse momentum $p_T > 60$ GeV and pseudorapidity $|\eta| < 2.8$,

- At least one electron with $p_T > 25$ GeV and $|\eta| < 2.47$, except for $1.37 < |\eta| < 1.52$,

- At least one muon with $p_T > 25$ GeV and $|\eta| < 2.7$,

- At least one photon with $p_T > 25$ GeV and $|\eta| < 2.37$.

The subsequent object selection then follows in Ref. [15], meaning that individual objects are kept only if they pass the following requirements:

- Electron candidates satisfying $p_T > 28$ GeV and $|\eta| < 2.47$ are selected. In the region $1.37 < |\eta| < 1.52$, known as the LAr crack region, electrons are rejected in order to reduce the contribution from non-prompt and fake electrons due to detector design in the liquid Argon calorimeter.

- Muon candidates are required to pass the Medium quality working point, with $p_T > 28$ GeV, and $|\eta| < 2.5$.

- Jet candidates satisfying $p_T > 25$ GeV and $|\eta| < 2.5$ are selected.

Finally, to remove as much background as possible with respect to the signal events, we define a signal region [15] that requires at least six jets, at least two of which are b-tagged, $H_T > 500$ GeV, and two leptons of the same sign, or at least three leptons for each event. The resulting efficiencies are shown in Table 1.

### 2.3 Data format

The generated Monte Carlo data were saved as ROOT files and then processed into CSV files using the event selection presented in Section 2.2. Each line in the CSV files is of variable length and contains three event specifiers followed by the kinematic features for each object in the event. The specific line format follows the event format used in the Dark Machines challenges [14, 16], given by

$$\text{event ID; process ID; weight; } \not{E}_T; \ \phi_{\not{E}_T}; \ \text{obj}_1, E_1, p_{T_1}, \eta_1, \phi_1; \ \text{obj}_2, E_2, p_{T_2}, \eta_2, \phi_2; ...$$

such that each object is represented by a string that starts with an identifier obj_n [2], followed by its kinematic properties in the form of a four-vector containing the full energy $E$ and the transverse momentum $p_T$ in units of MeV, as well as the pseudo-rapidity $\eta$ and the azimuthal angle $\phi$. The other relevant quantities are $\not{E}_T$ and $\phi_{\not{E}_T}$, which represent the magnitude of $E_T^{\text{miss}}$ and the azimuthal angle $\phi_{E_T^{\text{miss}}}$ of the missing transverse energy. The other three components represent the identity of an event, the corresponding physical process, and the event weight, which is given by the cross-section of the process divided by the total number of events generated.

Since the length of the events is variable, the data is zero-padded to the largest number of objects found in the events within in the entire dataset. The dataset includes 302 072 events,

---

[2]j: jet, b: b-jet, e-: electron, e+: positron, $\mu$-: muon, $\mu$+: antimuon, g: photon.

114  half of which correspond to the four tops signal and half of which are background processes.
115  All background processes have an equal number of events. Finally, the dataset was split in 80%
116  used for training, 10% used for validation and 10% used for testing. The data are available in
117  CSV format in Ref. [17].

## 3    Machine learning models

119  In this section we provide a brief summary of the models and methods used in this work.

### 3.1    Boosted decision tree

121  We use Light Gradient Boosting Machine (LightGBM[3]) for this study to test the performance
122  of Boosted Decision Tree (BDT).

123      BDTs combine a series of weak classifiers (decision trees) into a stronger classifier through
124  gradient boosting. The boosting strategy is defined with respect to a series of previous deci-
125  sion trees $f_1, f_2, ..., f_{t-1}$ which remain fixed, while the t-th tree $f_t$ is calculated. This process is
126  made highly efficient in LightGBM by converting the input data to histograms, and using gra-
127  dient based sampling to focus on the data that are not well modelled. This procedure reduces
128  memory usage and is optimized for both CPU and GPU performance. LightGBM uses first- and
129  second-order derivatives to minimize the loss for the next iteration for gradient boosting

$$\textbf{Loss}^{(t)} = \sum_{i=1}^{n} l(y_i, (\hat{y}_i^{(t-1)} + f_t(x_i))) + \sum_{i=1}^{t} \omega(f_i)$$
$$\approx \sum_{i=1}^{n} [g_i f_t(x_i) + \frac{1}{2} h_i f_t^2(x_i)] + \omega(f_t) + \textbf{constant} \tag{1}$$
$$g_i = \partial_{\hat{y}_i^{(t-1)}} l(y_i, \hat{y}_i^{(t-1)}), \quad h_i = \partial_{\hat{y}_i^{(t-1)}}^2 l(y_i, \hat{y}_i^{(t-1)}).$$

130  where $l$ is a reconstruction loss functions, e.g. Mean Square Error, Binary Cross Entropy, etc.,
131  $f_t$ is the t-th tree, and $\hat{y}^{(t-1)}$ is the class label predicted by $f_1, f_2, ..., f_{t-1}$. The term $\omega(f_i)$
132  represents tree complexity terms that involve properties such as depth, number of leaves, etc.
133  LightGBM uses a depth-first algorithm to add branches to the tree $f_t$ with limitation on largest
134  depth while minimizing Equation (1).

135      A common property of collision data in particle physics is a wide variability in object num-
136  bers and types. A structured formatting of the data would thus lead to a large degree of
137  sparsity, which we found to significantly degrade the BDT performance. We thus choose to
138  pre-process the data by limiting the maximum number of (jets, b-jets, $e^-$, $e^+$, $\mu^-$, $\mu^+$) to the
139  $(4, 4, 1, 1, 1, 1)$ hardest objects respectively.

140      One useful feature of this BDT is the fact that training is much faster than is the case for
141  all the other architectures described below. Therefore, fine-tuning of the hyperparameters and
142  adjustment of the data format is easy and efficient. Another attractive advantage of BDTs is
143  their capacity to indicate feature importance, which details which input value is most impor-
144  tant for its performance. In Section 4.1 we discuss the inclusion of high-level features beyond
145  the raw four-vectors, where this feature is especially useful.

---

[3]http://github.com/microsoft/LightGBM with `binary cross entropy` as loss function, `auc` as early stop
metric, 5000 estimators, 500 leaves, 0.01 learning rate, `gbdt` boost type, and max depth equaling to 15.

## 3.2 Fully-connected network

Fully-connected neural networks (FCNs) [18] are deep neural networks in their most basic form. They consist of several layers of neurons, each of which is connected to every neuron in the following layer. The connections represent a linear transformation with trainable parameters, which are followed by a non-linear activation function. After the final layer with a single node, a sigmoid activation is applied to produce a classification score.

The hidden layers all use ReLU activations, and the first five layers were followed by a Dropout layer with probability **0.5**. The network was trained with the Adam optimizer [19] with default parameters. An exponential learning rate schedule with $\gamma = 0.95$ was used, as well as early stopping by monitoring the performance on the validation set. The batch size and learning rates, as well as the network parameters were optimized using Optuna [20].

As is the case for BDTs, we found that the FCN performance generally deteriorates when applied to large, sparse input data. The data is thus pre-processed following the prescription given in Section 3.1.

## 3.3 Convolutional network

Convolutional Neural Networks (CNNs) [21] are mainly applied to analyze data where adjacent items have a causal relationship, i.e. image data. CNNs apply convolutional operations through trainable filter matrices that slide over the data to produce output that is translationally equivariant. The convolutional layers are usually followed by pooling operations to reduce the dimension of the data in the inner network layers. Here, we utilize a one-dimensional variant of such an architecture (1D CNN) called the DeepAK8 algorithm, originally used for jet tagging [22].

We incorporate 11 particle features as given in Table 2. Each feature is then represented by an array of size $N_{\mathrm{max}} = 18$, the maximum number of objects in an event. The event-wide features $E_T^{\mathbf{miss}}$ and $\phi_{E_T^{\mathrm{miss}}}$ are added to the $p_T$ and $\phi$ feature vectors respectively. Following [22], the network consists of a set of 1D convolutional blocks that pass over each of the feature vectors separately. The output of these blocks is concatenated and passed to a FCN with ReLU activations. The blocks are composed of two sub-blocks which consist of a set of convolutional layers with a ReLU activation function followed by a max pooling layer and a Dropout layer with dropout probability **0.2**. The model was trained with the Adam optimizer with default parameters, with a learning rate scheduler and early stopping which both monitor the validation AUC (Area Under the Curve) to prevent overfitting. The number of convolutional layers, the number of filters and the kernel size, as well as the FCN parameters were optimized with Optuna.

| Variables per particle |
|---|
| E, $p_T$, $\eta$, $\phi$, jet$_{\mathrm{tag}}$, b-jet$_{\mathrm{tag}}$, $e_{\mathrm{tag}}^-$, $e_{\mathrm{tag}}^+$, $\mu_{\mathrm{tag}}^-$, $\mu_{\mathrm{tag}}^+$, $\gamma_{\mathrm{tag}}$ |

Table 2: Particle input variables for the 1D CNN, Particle Net and Particle Transformer. FCNs and BDTs use the same variables, but limit the information of 4 momenta. For leptons only the lepton with the highet $p_T$ per lepton type (including charge) is considered, while for jets and $b-$jets, only the four jets with highest $p_T$ are considered. As a result, only 12 objects are used for FCNs and BDTs.

### 3.4  Particle Net

Particle Net (PN) [23] is a graph-based architecture based on Dynamic Graph Convolutional Neural Networks [24]. It treats events as particle cloud inspired by a point cloud [24] in Computer Vision challenges. Every final-state particle, encoded by the variables shown in Table 2, is represented by an individual node in the graph, carrying ($E$, $p_T$, $\eta$, $\phi$) as node values. Edges are constructed by connecting these particles with their k-nearest neighbours (kNN), where distances are defined as $\Delta R_{ij} = \sqrt{(\Delta \eta)^2_{ij} + (\Delta \phi)^2_{ij}}$. The graph representing the event thus has $N$ (number of final state particles) nodes and $kN$ edges.

Messages are passed to every node $i$ by all $k$ neighbouring nodes $j$ in the graph by applying the operation

$$x'_i = \frac{1}{k} \sum_{j=1}^{k} \text{FCN}(x_i, x_i - x_{i_j}) \tag{2}$$

to every node. Here, $j$ runs over the $k$ nearest neighbors of $i$ and the weights of the FCN are the same for every node and edges combination. We performed experiments with an attention-weighted procedure rather than the simple averaging over edges of Equation (2), but found no difference in performance.

The node features are then updated to $x'_i$. Multiple layers of the above procedure are applied consecutively, and the node features after every step are concatenated, averaged over the nodes, and then processed by another FCN which also receives $E_T^{\text{miss}}$ and $\phi_{E_T^{\text{miss}}}$ to obtain a classification.

We performed a wide hyperparameter scan over the ParticleNet architecture and found no significant difference in performance as long as sufficient capacity is available. We thus choose to use the hyperparameter settings recommended in Ref. [23] and the training procedure of [25]. Our implementation is based on Ref. [26].

### 3.5  Particle Transformer

Particle Transformer (ParT) [8] is a transformer-based architecture originally developed for jet tagging. It is inspired by the success of similar architectures in fields such as natural language processing [27], embedding individual particles rather than words. At its core lies the repeated application of the self-attention mechanism

$$\text{Attention}(Q, K, T) = \text{SoftMax}\left(QK^T / \sqrt{d}\right) V, \tag{3}$$

where $Q$, $K$ and $V$ are trainable $d$-dimensional linear projections of the particle embedding based on the variables of Table 2.

The application of the attention mechanism serves to correlate every particle with all others. Furthermore, it is applicable to vary numbers of particles and is explicitly permutation invariant. Classification is obtained by appending a classification token to the list of particle embeddings before the last few layers of the transformer. This token is a trainable set of weights that is identical for every event. The attention mechanism then correlates the classification token with the event, after which it is processed by an FCN, which also receives $E_T^{\text{miss}}$ and $\phi_{E_T^{\text{miss}}}$, to produce a classification label. Our implementation is based on Ref. [28].

As was the case for Particle Net, a hyperparameter scan over the ParT architecture does not lead to significant differences in performance. We thus choose to use the hyperparameter settings and training procedure recommended in Ref. [8].

### 3.6 Particle Transformer as Set Transformer

We incorporate the Set Transformer architecture [29] into the ParT model. In a Set Transformer, the matrix $Q$ in Eq. (3) is no longer a projection of the input states, but rather a set of so-called inducing points. These are parameters that are jointly optimized with the rest of the parameters of the transformer. The model should thus learn to use $Q$ to effectively summarize the information contained in $V$ for any possible state.

This modification leads to a self-attention mechanism that is permutation *invariant* [30], meaning that permuted inputs produce exactly the same output. The usual self-attention mechanism is permutation *equivariant*, meaning that the outputs permute along with the inputs. Since collision data presents as an unordered set of particles, the performance of the model may benefit from the former, as it imposes a stricter constraint.

In our experiments, we explored various configurations for the number of inducing points, specifically testing sets of {18, 20, 30, 40, 50, 100, 200} points. While the performance of the transformer model without pairwise features increased slightly with increasing number of inducing points, we did not observe any improvement with increasing number of inducing points in the model with pairwise features. Once again, the best Set Tranformer model proved to be the one containing all pairwise kinematic interactions as explained in the following chapter (labelled 'SetT$_{\text{int. SM}}$').

### 3.7 Particle Transformer with Focal Loss

For the particle transformer we perform experiments using the focal loss [31] in place of the usual cross-entropy loss. It is given by

$$\text{Focal Loss} = -\alpha_t (1 - p_t)^\gamma \log(p_t). \tag{4}$$

where:

- $\alpha_t$ is a balancing factor that weights the importance of the different classes $t$. The alpha parameter essentially adjusts the importance given to each class and can handle class imbalances.

- $p_t$ is the model's estimated probability for the class label $t$,

- $\gamma$ is the focusing parameter, which adjusts the rate at which easy-to-classify examples are down-weighted. High $\gamma$ values would decrease the contribution of events which are very much signal-like, i.e. $p_t$ were is large.

While the focal loss was originally developed to handle class imbalance, it can still enhance model performance in other cases. The scaling factor $(1 - p_t)^\gamma$ increases the weight of difficult training samples, where the model does not yet assign large probability, while attenuating the loss for well-classified samples.

We performed a comprehensive hyperparameter scan over the focal loss parameters using the extended ParT model (with SM running coupling constants including the pairwise kinematic features with the (third) SM interaction matrix as explained in the following chapter). Scans were performed over $\alpha \in \{0.25, 0.5, 0.75, 1\}$ and $\gamma \in \{0, 1, 2, 3, 4, 5, 6\}$. The best results were achieved for $\alpha = 0.75$ and $\gamma = 3$ (the model is labelled 'ParT$_{\text{int. SM (FL)}}$'). Even at these optimal values, the overall model performance was not better than that of models trained with the usual cross-entropy loss. Thus, results presented below pertain to models trained with cross-entropy loss, unless otherwise specified. However, this conclusion is highly dependent on the mixture of background processes. In particular, the focal loss leads to better performance in separating some backgrounds.

## 4  Informing the ML models about physics

### 4.1  Pairwise kinematic features based on 4-vectors

Previous work has highlighted that the inclusion of information beyond raw four-vector data, such as correlations of four-vectors (called here pairwise features), can improve deep learning classifier performance [8, 25] in jet physics. These pairwise 4-vector correlations are invariant masses or distances between two objects, which are typically known from jet physics.

Similarly, it is common practice to include high-level features in the training of BDTs to improve event classification, see e.g. [32]. The work of Ref. [25] suggests that this increase in performance is due to the resulting implicit embedding of Lorentz symmetry in the network architecture through features that adhere to (sub)symmetries. Lorentz's symmetry has previously been shown to function as a strong inductive bias for neural network design [33–36].

For the BDT, we perform experiments with the inclusion of a variety of high-level features, which are treated on the same footing as the low-level ones. Similarly, we follow [8, 25] and include pairwise features in Particle Net and Particle Transformer through a trainable embedding $U_{ij}$ for particles $i$ and $j$. They are then included in Particle Net by replacing Equation (2) with

$$x'_i = \frac{1}{k} \sum_{j=1}^{k} \text{FCN}(x_i, x_i - x_{i_j} + U_{ij}) \tag{5}$$

and in Particle Transformer by replacing Equation (3) with

$$\text{Attention}(Q, K, T) = \text{SoftMax}\left(QK^T/\sqrt{d} + U\right)V. \tag{6}$$

In all three above cases, we evaluated the performance of a wide variety of kinematic pairwise features, including $m_{ij}$, $\Delta R_{ij}$, the jet-based features used in Ref. [8] and three-body invariant masses. Using the feature importance indicator of the BDT, and empirically for Particle Net and Particle Transformer, we find that for all architectures the performance is saturated by the inclusion of only $m_{ij}$ and $\Delta R_{ij}$. Furthermore, the BDT indicates that we find that the pairwise invariant masses lead to the biggest gain in performance. This result is in line with the findings of [25]. In the next section, we experiment with adding further information through dynamics, while maintaining the above kinematic information.

### 4.2  Pairwise kinematic features and the Standard Model Interaction Matrix

The Standard Model (SM) of particle physics provides the most comprehensive framework for understanding the electromagnetic, weak and strong nuclear interactions between elementary particles. We explore incorporating the dynamics of particle interactions described by the SM through the inclusion of a separate interaction matrix in the embedding $U_{ij}$ for the PN and ParT models. The interaction matrix consists of entries indicating the significance of pairwise particle interactions.

To systematically investigate the effect of adding dynamic information to the models, we explore the use of three types of interaction matrices with increasing amounts of physical information. In the first matrix (abbreviation SMids and called SM matrix[1] in Table 3 ), an entry '1' indicates an interaction possible at leading order in the SM, while a '0' indicates interactions that only appear at higher orders. The following pairwise interactions are assigned a '1' in the matrix: jet–jet, jet–b-jet, jet–$\gamma$, b-jet–b-jet, b-jet–$\gamma$, $e^- - e^+$, $e^- - \gamma$, $e^+ - \gamma$, $\mu^- - \mu^+$, $\mu^- - \gamma$. $\mu^+ - \gamma$. The omission of other particle interactions with a "0" does not mean that they are physically impossible, but is a practical limitation for the model. The simplified representation does not take into account the full complexity of the SM, but should provide a computationally tractable method for learning high-level interaction features.

In the second iteration of the interaction matrix (abbreviation SMconst and called SM matrix[2] in Table 3) we use the coupling constants of the SM as fixed parameters: $g_Z = 0.758$ for the weak force for leptons, $g_s = 1.22$ for the strong force in jet interactions, and $g_e = 0.31$ for the electromagnetic force in photon interactions. The interactions between jets and photons as well as between b-jets and photons are determined by the electromagnetic coupling constant $g_e$, since photons have no colour charge. Consequently, the interactions are characterized as follows:

- For the jet-$\gamma$ interactions, the modified coupling constant is $g_e \times 0.5$ to reflect the assumption that jets originate mainly from quarks for the signals investigated in this work. The factor $0.5$ in the jet-$\gamma$ coupling constant comes from the average charge of the quarks, which is calculated as $\left(\frac{1}{3} + \frac{2}{3}\right)/2$, assuming an equal distribution of the quark charges of $\frac{1}{3}$ and $\frac{2}{3}$.

- For the b-jet-$\gamma$ interactions, we take into account the electric charge of the b-quarks by using $g_e \times \frac{1}{3}$.

For the third interaction matrix (abbreviation SM and called SM matrix[3] in Table 3) we take the energy dependence of the coupling constants into account:

- For QED, the running of the fine-structure constant $\alpha$ is described by the Renormalization Group Equation (RGE). At one-loop level for a given pair of particle types $(i, j)$, it can be approximated as:

$$\alpha(Q^2) = \frac{\alpha(\mu_0^2)}{1 - \frac{n\alpha(\mu_0^2)}{3\pi} \cdot \ln\left(\frac{Q^2}{\mu_0^2}\right)},$$

$$g_e = \sqrt{4\pi\alpha} \tag{7}$$

  The factor $n$ approximates the contribution of the different particles in the loop. We used $n = 3$ and considered only leptons. Other choices did not have much influence.

- For QCD, the running of $\alpha_s$ is more complex due to the non-Abelian nature of the theory. The one-loop RGE for a given pair of particle types $(i, j)$, $\alpha_s$ is:

$$\alpha_s(Q^2) = \frac{\alpha_s(\mu_0^2)}{1 + \frac{\alpha_s(\mu_0^2)(33-2n_f)}{12\pi} \ln\left(\frac{Q^2}{\mu_0^2}\right)},$$

$$g_s = \sqrt{4\pi\alpha_s} \tag{8}$$

  Here $\mu_0 = 91.1876$ GeV, $\alpha(\mu_0) = \frac{1}{127.5}$, $\alpha_s(\mu_0) = 0.118$, $n_f = 6$ is the number of quark flavors that are active at the energy scale $Q^2$. To calculate these constants at a specific scale, such as the average transverse momentum $\bar{p}_t$ of a particle pair in an event, we set $Q^2 = \bar{p}_t^2 = \left(\frac{p_t^i + p_t^j}{2}\right)^2$ as the energy scale in the RGEs to calculate $\alpha(Q^2)$ and $\alpha_s(Q^2)$.

  $g_e$ gives the effective coupling strength for electromagnetic interactions, while $g_s$ gives the effective coupling strength for strong interactions at a given energy scale $Q^2$. We used $g_z$ as a constant value from the previous version of the matrix.

The interaction matrix provides a structured approach to encode SM-particle interactions for training machine learning models, especially models such as ParT. By simplifying the wide range of possible interactions into a prioritized scheme, the matrix allows learning to focus on the most important interactions. The interaction matrices here are structured in such a way that large negative numbers (-10k) are used if no particle exist (i.e. masked). This masking is achieved by the softmax activation function, which exponentiates the values in the attention matrix and thus pushes the irrelevant values towards zero.

# 5 Results

## 5.1 Summary of Model details

Table 3 summarizes the details of the ML models and the sessions that we found after the hyperparameter studies discussed above. In the following sections, we will discuss the performance of these models applied to 4 top signals with different backgrounds and to signals with top-top-Higgs events.

| NN structure | Pairwise kinematic features | Loss function |
|---|---|---|
| BDT | | |
| BDT$_{int.}$ | $m_{ij}, \Delta R_{ij}$ | |
| FCN | | |
| CNN | | |
| PN | | Cross-entropy |
| PN$_{int.}$ | $m_{ij}, \Delta R_{ij}$ | |
| PN$_{int.\,SMids}$ | $m_{ij}, \Delta R_{ij}$ + SM matrix[1] | |
| PN$_{int.\,SM\,const}$ | $m_{ij}, \Delta R_{ij}$ + SM matrix[2] | |
| PN$_{int.\,SM}$ | $m_{ij}, \Delta R_{ij}$ + SM matrix[3] | |
| ParT | | |
| ParT$_{int.}$ | $m_{ij}, \Delta R_{ij}$ | |
| ParT$_{int.\,SM\,(FL)}$ | $m_{ij}, \Delta R_{ij}$ + SM matrix[3] | Focal $[\alpha = 0.75, \gamma = 3]$ |
| ParT$_{int.\,SMids}$ | $m_{ij}, \Delta R_{ij}$ + SM matrix[1] | |
| ParT$_{int.\,SM\,const}$ | $m_{ij}, \Delta R_{ij}$ + SM matrix[2] | |
| ParT$_{int.\,SM}$ | $m_{ij}, \Delta R_{ij}$ + SM matrix[3] | Cross-entropy |
| SetT$_{int.\,SM}$ | $m_{ij}, \Delta R_{ij}$ + SM matrix[3] | |

Table 3: Summary of Machine Learning (ML) model details, including neural network (NN) structures and their respective loss functions. This table also highlights the inclusion of pairwise kinematic features in certain models. The particle input variables for these models are detailed in Table 2.

## 5.2 A search for 4 top production

In order to investigate the relationship between the amount of training data and the model's performance, we plotted learning curves on the Fig. 1 that shows the area under the ROC curve (AUC) scores as a function of training size. The x-axis represents the size of the training set, while the y-axis denotes the AUC score achieved by the model on a test set.

As illustrated in the figure, there is a clear trend of improving AUC scores with an increase in the training set size, affirming the hypothesis that larger datasets enhance model performance. Notably, this improvement is more pronounced in the initial stages of increasing the data

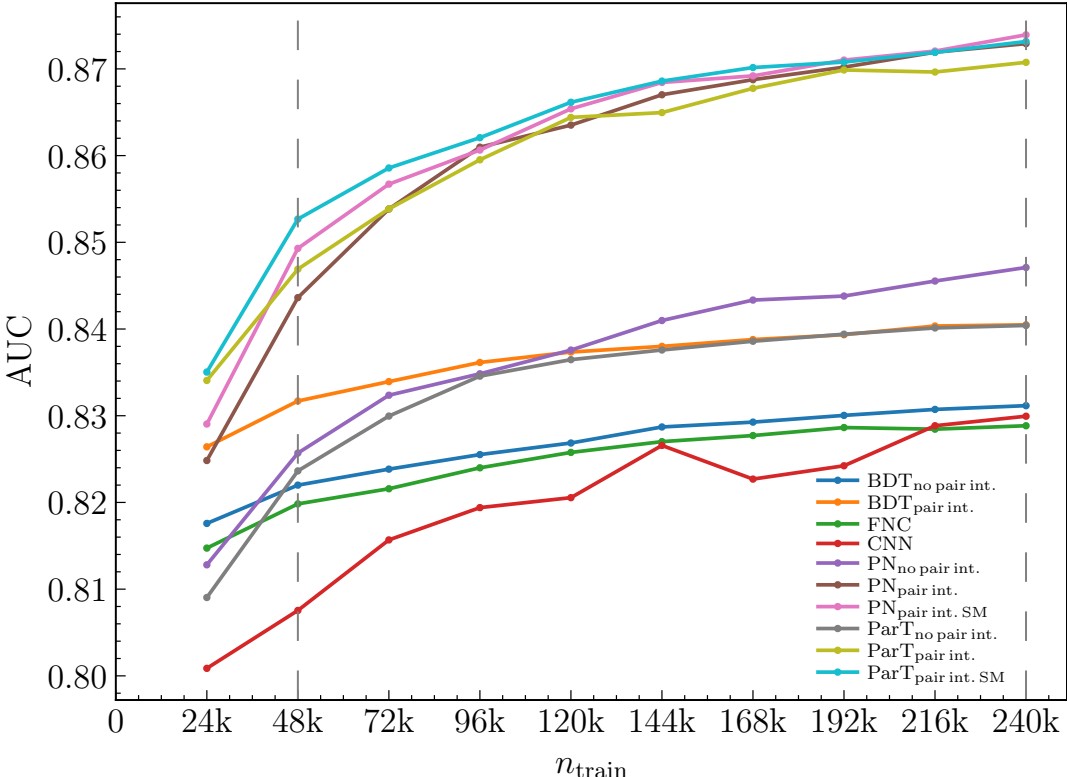

Figure 1: Learning curves of various machine learning models. This plot illustrates the relationship between the size of the training dataset and the AUC (Area Under the Curve) for each model. The vertical dashed lines represent the referenced training sizes of 48k and 240k event in the dataset.

volume. Beyond a certain point, however, the rate of improvement in AUC scores begins to plateau. This observation suggests that while additional training data is beneficial, the marginal gains in model accuracy diminish after reaching a certain dataset size.

Furthermore, the learning curves also provide insights into the data efficiency and learning capacity of the different models. Models like PN and ParT demonstrate a steeper ascent in the AUC scores with fewer data, indicating better data efficiency, while others show a more gradual improvement, reflecting their need for larger datasets to achieve comparable performance.

Our analysis and its key findings, based on the 48k training dataset, are summarized in Table 4. Full details covering the entire 240k training dataset can be found in Table 8 in the Appendix. This table shows the AUC values together with the background efficiencies ($\epsilon_B$) at signal efficiencies ($\epsilon_S$)[4] of 30% and 70% for each evaluated method. In particular, the PN and ParT architectures with the inclusion of pairwise features and SM running coupling constants (labelled 'int. SM') consistently achieved the best performance in all metrics and at different signal efficiencies.

The background efficiencies at a signal efficiency of **30**% vary among the models. Generally, a lower background efficiency at this signal efficiency level indicates a model's strength in maintaining signal detection while effectively rejecting a significant portion of the background. At a higher signal efficiency of **70**%, the background efficiencies increase for all models, which is expected as increasing signal efficiency typically comes at the cost of allowing more back-

---

[4] $\epsilon_S \equiv \frac{\text{TP}}{\text{TP+FN}}$ and $\epsilon_B \equiv \frac{\text{FP}}{\text{TN+FP}}$.

| | | BDT | BDT$_{\text{int.}}$ | FCN | CNN |
|---|---|---|---|---|---|
| $t\bar{t}+h$ | AUC | 0.825(0) | 0.831(0) | 0.821(2) | 0.778(6) |
| | $\epsilon_B(\epsilon_S = 0.7)$ | 0.206(0) | 0.192(0) | 0.203(1) | 0.272(11) |
| | $\epsilon_B(\epsilon_S = 0.3)$ | 0.026(1) | 0.026(0) | 0.026(1) | 0.037(1) |
| $t\bar{t}+W$ | AUC | 0.891(0) | 0.895(0) | 0.887(0) | 0.867(5) |
| | $\epsilon_B(\epsilon_S = 0.7)$ | 0.099(0) | 0.092(0) | 0.103(1) | 0.125(8) |
| | $\epsilon_B(\epsilon_S = 0.3)$ | 0.011(0) | 0.011(0) | 0.010(0) | 0.011(1) |
| $t\bar{t}+WW$ | AUC | 0.740(0) | 0.746(0) | 0.737(1) | 0.745(2) |
| | $\epsilon_B(\epsilon_S = 0.7)$ | 0.347(0) | 0.339(0) | 0.342(5) | 0.335(3) |
| | $\epsilon_B(\epsilon_S = 0.3)$ | 0.050(0) | 0.051(0) | 0.054(0) | 0.051(0) |
| $t\bar{t}+Z$ | AUC | 0.833(0) | 0.856(0) | 0.836(0) | 0.839(1) |
| | $\epsilon_B(\epsilon_S = 0.7)$ | 0.191(0) | 0.163(0) | 0.192(0) | 0.190(4) |
| | $\epsilon_B(\epsilon_S = 0.3)$ | 0.026(0) | 0.019(0) | 0.023(0) | 0.021(1) |
| | | PN | PN$_{\text{int.}}$ | PN$_{\text{int. SM}}$ | ParT$_{\text{int. SM (FL)}}$ |
| $t\bar{t}+h$ | AUC | 0.824(0) | 0.842(1) | **0.846(1)** | 0.844(1) |
| | $\epsilon_B(\epsilon_S = 0.7)$ | 0.199(0) | 0.176(3) | **0.171(2)** | 0.176(2) |
| | $\epsilon_B(\epsilon_S = 0.3)$ | 0.025(0) | **0.019(1)** | 0.020(1) | 0.020(1) |
| $t\bar{t}+W$ | AUC | 0.887(0) | 0.895(2) | 0.900(1) | **0.902(4)** |
| | $\epsilon_B(\epsilon_S = 0.7)$ | 0.102(1) | 0.097(1) | **0.091(1)** | **0.091(5)** |
| | $\epsilon_B(\epsilon_S = 0.3)$ | 0.011(0) | 0.011(0) | **0.010(0)** | 0.011(0) |
| $t\bar{t}+WW$ | AUC | 0.742(0) | 0.760(1) | 0.765(0) | 0.768(3) |
| | $\epsilon_B(\epsilon_S = 0.7)$ | 0.335(2) | 0.311(1) | 0.297(2) | 0.294(7) |
| | $\epsilon_B(\epsilon_S = 0.3)$ | 0.051(0) | 0.044(1) | **0.044(1)** | 0.044(1) |
| $t\bar{t}+Z$ | AUC | 0.851(0) | 0.879(1) | **0.887(1)** | 0.892(0) |
| | $\epsilon_B(\epsilon_S = 0.7)$ | 0.168(4) | 0.136(1) | **0.126(2)** | 0.119(4) |
| | $\epsilon_B(\epsilon_S = 0.3)$ | 0.020(0) | 0.016(1) | 0.016(0) | 0.016(0) |
| | | ParT | ParT$_{\text{int.}}$ | ParT$_{\text{int. SM}}$ | SetT$_{\text{int. SM}}$ |
| $t\bar{t}+h$ | AUC | 0.824(0) | 0.837(2) | **0.846(1)** | 0.845(1) |
| | $\epsilon_B(\epsilon_S = 0.7)$ | 0.197(3) | 0.179(6) | 0.174(1) | 0.176(3) |
| | $\epsilon_B(\epsilon_S = 0.3)$ | 0.023(0) | 0.020(0) | 0.020(0) | 0.020(0) |
| $t\bar{t}+W$ | AUC | 0.896(1) | 0.899(1) | **0.905(2)** | 0.898(1) |
| | $\epsilon_B(\epsilon_S = 0.7)$ | 0.097(2) | 0.090(1) | **0.089(3)** | 0.094(2) |
| | $\epsilon_B(\epsilon_S = 0.3)$ | 0.010(0) | 0.010(0) | **0.009(0)** | 0.011(0) |
| $t\bar{t}+WW$ | AUC | 0.737(0) | 0.767(1) | **0.769(0)** | 0.763(1) |
| | $\epsilon_B(\epsilon_S = 0.7)$ | 0.354(3) | 0.295(5) | **0.288(2)** | 0.301(5) |
| | $\epsilon_B(\epsilon_S = 0.3)$ | 0.050(1) | 0.040(0) | **0.042(0)** | 0.047(1) |
| $t\bar{t}+Z$ | AUC | 0.839(1) | 0.885(0) | **0.891(1)** | 0.886(2) |
| | $\epsilon_B(\epsilon_S = 0.7)$ | 0.182(2) | 0.130(1) | **0.119(3)** | 0.129(4) |
| | $\epsilon_B(\epsilon_S = 0.3)$ | 0.021(1) | 0.016(0) | **0.015(0)** | **0.014(0)** |

Table 4: The areas under the ROC curve and the background efficiencies, at signal efficiencies of **70%** and **30%** respectively, correspond to the 48k training dataset. Quoted uncertainties are extracted from three independent runs for each network architecture. Numbers in bold indicate the best performance. In cases where the performances of multiple architectures are the best within the uncertainty, the results are both indicated.

ground events. Certain models, particularly PN$_{\text{int. SM}}$ and ParT$_{\text{int. SM}}$ with pairwise kinematic features and the SM interaction matrix, manage to maintain relatively lower background effi-

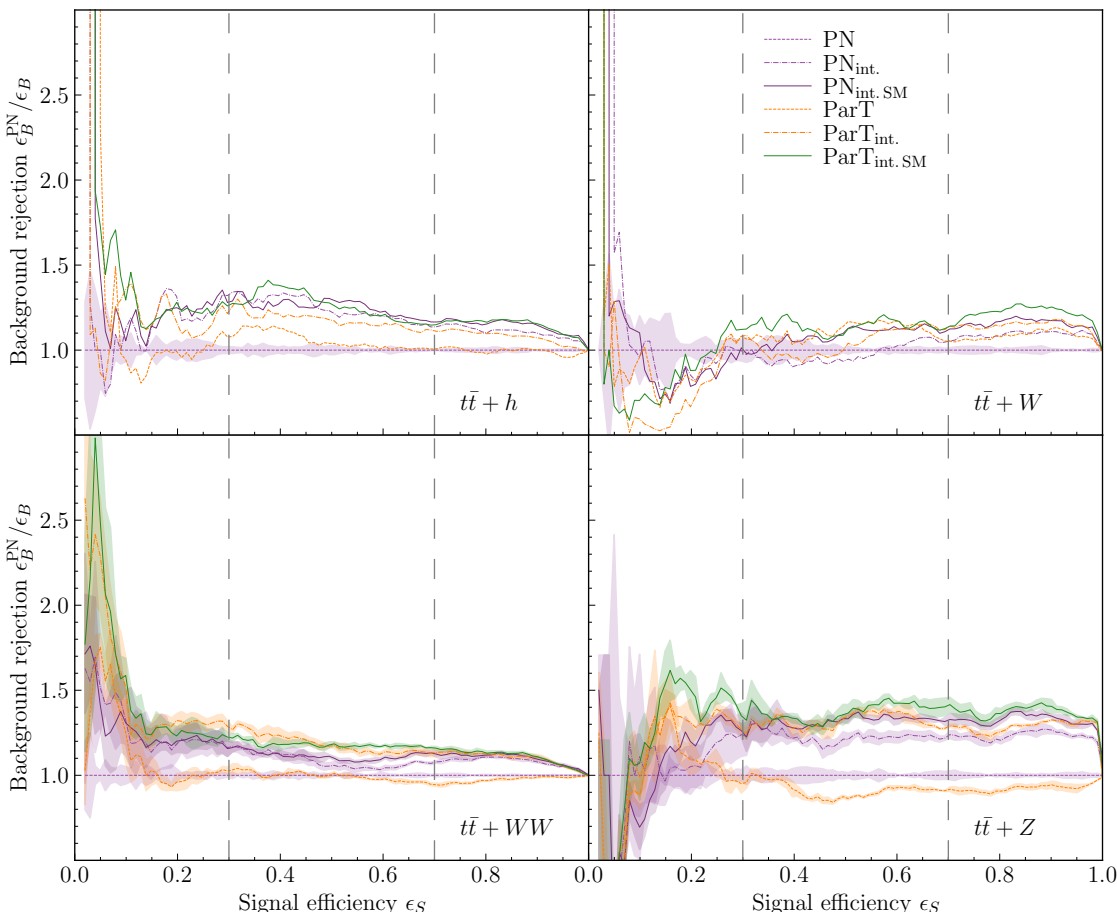

Figure 2: Signal efficiency versus background rejection plot for the four background processes corresponding to the 48k training dataset.

ciencies, underscoring their efficiency in handling a more challenging balance between signal and background.

Figure 2 presents an alternative metric that more effectively illustrates the significance of these differences. In this comparison, we evaluate the background rejection as a function of signal efficiencies to a PN baseline model using the 48k training dataset. The PN and ParT architectures including physical information, particularly $PN_{int. SM}$ and $ParT_{int. SM}$ with pairwise kinematic features and the SM interaction matrix, demonstrate an improvement in background reduction compared to the PN basedline model of 10-40% for signal efficiencies between 30 and 90%. Best performance is found for models that include the SM interaction matrix.

Fig. 3 shows the signal and background distributions as a function of the classifier score, normalized to the total cross-section. This figure, with its solid lines and error bands, contains the mean and standard deviation observed over three independent runs for each architecture across the entire dataset. A critical observation here is the tendency of the best performing architectures to concentrate large portions of the background at lower classifier values, especially for background processes with higher cross-sections such as $t\bar{t} + W$ and $t\bar{t} + Z$. This property is of crucial importance for the discrimination of backgrounds in signal fits in LHC experiments.

Table 5 compares the performance of various models at two distinct signal efficiency levels, $\epsilon_S = 0.3$ and $\epsilon_S = 0.7$. Significance, denoted as $\sigma$, is defined as the signal count $s$ divided by the square root of the background count $b$. This calculation is done under the assumption of

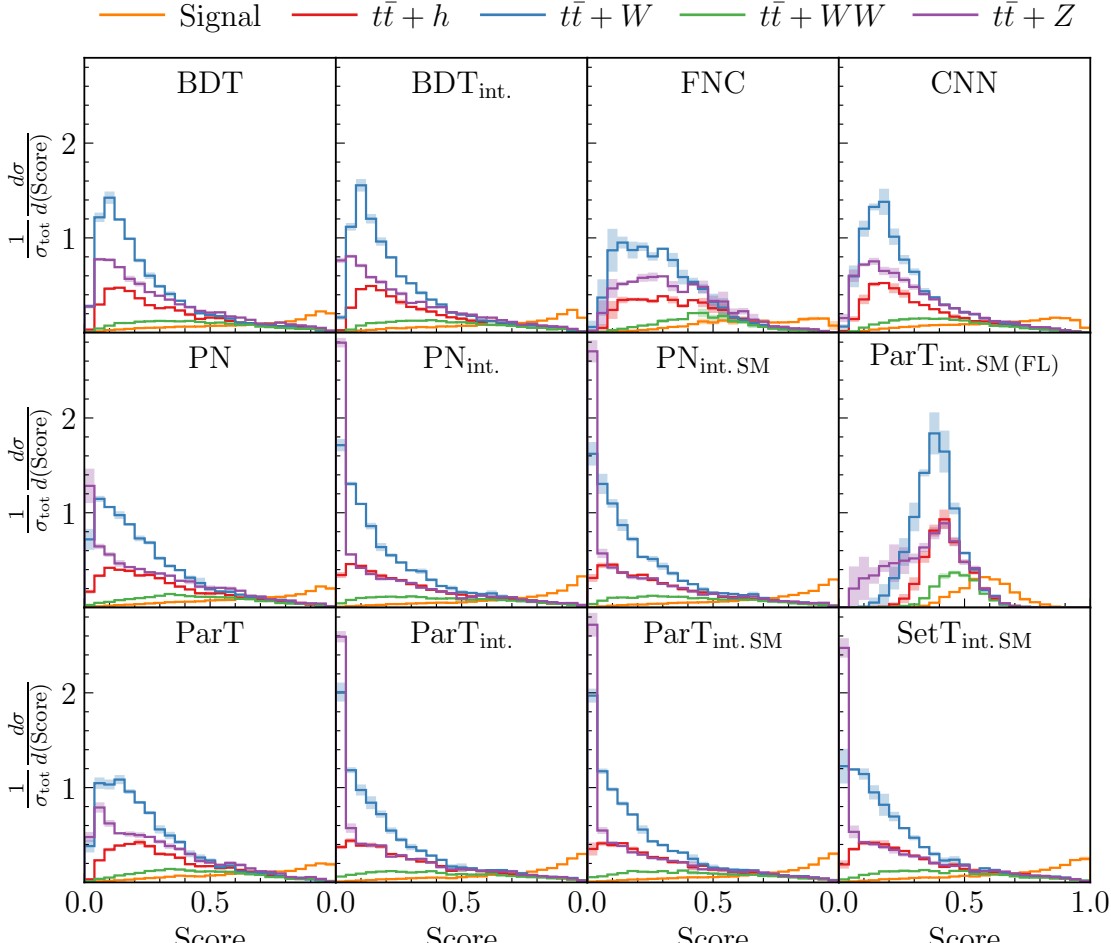

Figure 3: The distribution of the signal and the backgrounds as a function of the classifier score, normalized to the total cross-section. The solid lines and error bands correspond to the mean and standard deviation over three independent runs for each architecture across the entire dataset.

a luminosity of $100 \, \text{fb}^{-1}$ and incorporates the LO (Leading Order) cross-section from Table 1.
We also consider the impact of systematic errors on the significance, represented by $\sigma_{\delta_{\text{sys}}=0.2}$,
which is calculated as $s$ divided by the square root of the effective background count $b_{\text{sys}}$,
where $b_{\text{sys}} = b + (b \cdot \delta_{\text{sys}})^2$ and $\delta_{\text{sys}}$ is set to $0.2$.

At $\epsilon_S = 0.3$, the model captures **30%** of the true signal events. The significance without
systematic errors at this level suggests effective discrimination between signal and background.
However, introducing a systematic error of **20%** noticeably reduces the significance, under-
scoring the influence of factors like instrumental or theoretical uncertainties. At $\epsilon_S = 0.7$,
the model identifies **70%** of the true signal events. While this higher efficiency captures more
signal events, the corresponding raw significance drops. The impact of systematic errors is
more pronounced at this efficiency, as evidenced by a further decrease in $\sigma_{\delta_{\text{sys}}=0.2}$.

|  |  | $\sigma$ | $\sigma_{\delta sys = 0.2}$ |
|---|---|---|---|
| BDT | $\epsilon_S = 0.3$ | 20.77 | 6.79 |
|  | $\epsilon_S = 0.7$ | 16.82 | 2.01 |
| BDT$_{\text{int.}}$ | $\epsilon_S = 0.3$ | 21.93 | 7.53 |
|  | $\epsilon_S = 0.7$ | 17.51 | 2.17 |
| FCN | $\epsilon_S = 0.3$ | 20.31 | 6.51 |
|  | $\epsilon_S = 0.7$ | 16.67 | 1.97 |
| CNN | $\epsilon_S = 0.3$ | 20.88 | 6.86 |
|  | $\epsilon_S = 0.7$ | 16.73 | 1.98 |
| PN | $\epsilon_S = 0.3$ | 23.09 | 8.29 |
|  | $\epsilon_S = 0.7$ | 17.68 | 2.21 |
| PN$_{\text{int.}}$ | $\epsilon_S = 0.3$ | 25.30 | 9.83 |
|  | $\epsilon_S = 0.7$ | **20.51** | **2.97** |
| PN$_{\text{int. SM}}$ | $\epsilon_S = 0.3$ | **25.65** | **10.09** |
|  | $\epsilon_S = 0.7$ | **20.50** | **2.97** |
| ParT | $\epsilon_S = 0.3$ | 22.37 | 7.82 |
|  | $\epsilon_S = 0.7$ | 17.72 | 2.23 |
| ParT$_{\text{int.}}$ | $\epsilon_S = 0.3$ | 24.54 | 9.29 |
|  | $\epsilon_S = 0.7$ | 20.21 | 2.89 |
| ParT$_{\text{int. SM}}$ | $\epsilon_S = 0.3$ | 25.36 | 9.88 |
|  | $\epsilon_S = 0.7$ | **20.53** | **2.98** |
| ParT$_{\text{int. SM (FL)}}$ | $\epsilon_S = 0.3$ | **26.19** | **10.48** |
|  | $\epsilon_S = 0.7$ | 20.28 | 2.91 |
| SetT$_{\text{int. SM}}$ | $\epsilon_S = 0.3$ | **25.58** | **10.03** |
|  | $\epsilon_S = 0.7$ | 20.18 | 2.88 |

Table 5: Significance table calculated for the entire dataset.

Comparative analysis reveals that the different versions of the particle transformer with
SM interaction matrix (PartT$_{\text{int SM}}$ with and without focal loss and as Set Transformer) achieve
the highest significance without systematic errors at $\epsilon_S = 0.3$. In addition, ParT$_{\text{int. SM}}$ attains
the highest significance, accounting for systematic errors at $\epsilon_S = 0.7$. These findings highlight
the crucial role of model selection based on specific analytical requirements and the significant
impact of systematic errors, especially at higher signal efficiency levels.

Compared to the baseline graph network (PN), it is interesting to estimate how much the
sample statistic (or integrated luminosity) would have to be increased in order to achieve a
similar increase in significance, neglecting systematic errors. An increase of significance e.g.
from **2.21** $\sigma$ (baseline PN model at 70% signal efficiency) to **2.98** $\sigma$ (ParT$_{\text{int. SM}}$) corresponds

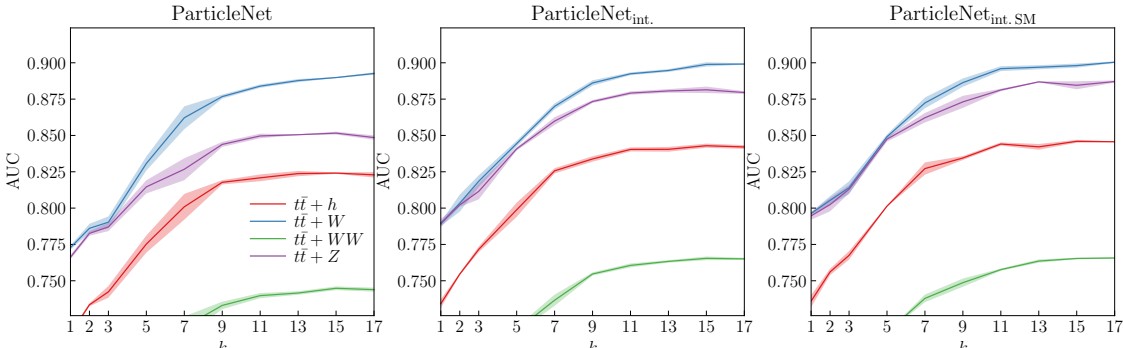

Figure 4: The performance of Particle Net on the four background processes as a function of $k$, the number of nearest neighbours. The solid lines and error bands correspond to the mean and standard deviation over three independent runs for every value of $k$ correspond to the 48k training dataset.

to an increase in integrated luminosity of approximately 82%. An increase of significance e.g. from **8.29 $\sigma$** (baseline PN model at 30% signal efficiency) to **9.88 $\sigma$** (ParT$_{int. SM}$) corresponds to an increase in integrated luminosity of approximately 42% and an increase from **8.29 $\sigma$** to **10.48 $\sigma$** (ParT$_{int. SM (FL)}$) corresponds to an increase in integrated luminosity of approximately 60%.

Finally, in Fig. 4 shows the performance corresponding to the 48k training dataset using the AUC metric for PN as a function of k, the number of nearest neighbors. As one might expect, performance improves with k, eventually saturating when k approaches the limit where every particle is connected to all other particles. Note that this limit would require more significant computational overhead in the context of jet physics, as the number of objects can grow much larger and the complexity scales like $\mathcal{O}(kn)$. Here, the number of objects is limited and setting k = $n$ is unproblematic. Note that this setup converts PN into an architecture that is quite similar to ParT, as is reflected in the results.

## 5.3  The top-top-Higgs searches

To evaluate the impact of including ongoing coupling constants on the efficiency of neural networks, a second analysis was performed focusing on the search for top-top-Higgs signals. The main results of this study are presented in Table 6. The table shows the AUC for both 4 top and top-top-Higgs signal detection. The first row of the table shows the AUC values obtained for the 4 top signal, followed by the second row explaining the results for the top-top-Higgs signal. It is important to emphasize that the entire data set was used when analysing the 4 top signal, while the data set for the top-top-Higgs signal, although identical in composition, intentionally excluded the data of the 4 top signal. This study shows three different ML models: the standard ParT architecture, the extended ParT with integrated pairwise features (labelled 'int.') and the extended ParT with SM running coupling constants (labelled 'int. SM'). The model containing both the pairwise features and the SM interaction matrix performs best, which again confirms our earlier results. Here too, the background can be significantly reduced by about 30% compared to a PN baseline model.

Table 7 shows the significance estimate for the simplified top-top Higgs analysis. Here we have not included all backgrounds (compared to a full ATLAS or CMS analysis), and the values should only be used to see how important background reduction can be. For example, an increase in significance from **3.80 $\sigma$** (base PN model at 70% signal efficiency) to **5.02 $\sigma$** (PN$_{int. SM}$) corresponds to an increase in integrated luminosity of about 75%, which confirms

⁴⁴⁹ our previous results.

⁴⁵⁰     Comprehensive results covering other versions of the SM interaction matrices are presented
⁴⁵¹ in Table 9 in the Appendix.

|  |  | PN | PN$_{\text{int.}}$ | PN$_{\text{int. SM}}$ |
|---|---|---|---|---|
| | AUC | 0.8471(1) | 0.8729(0) | **0.8739(0)** |
| $t\bar{t}t\bar{t}$ | $\epsilon_B(\epsilon_S = 0.7)$ | 0.1758(3) | 0.1387(1) | **0.1369(1)** |
| | $\epsilon_B(\epsilon_S = 0.3)$ | 0.0207(0) | 0.0182(0) | **0.0176(0)** |
| | AUC | 0.8146(2) | 0.8505(0) | **0.8523(0)** |
| $t\bar{t}+h$ | $\epsilon_B(\epsilon_S = 0.7)$ | 0.2292(1) | 0.1787(0) | **0.1733(1)** |
| | $\epsilon_B(\epsilon_S = 0.3)$ | 0.0471(1) | 0.0345(0) | **0.0340(0)** |
| | | ParT | ParT$_{\text{int.}}$ | ParT$_{\text{int. SM}}$ |
| | AUC | 0.8404(0) | 0.8708(0) | **0.8732(0)** |
| $t\bar{t}t\bar{t}$ | $\epsilon_B(\epsilon_S = 0.7)$ | 0.1842(3) | 0.1394(0) | **0.1366(0)** |
| | $\epsilon_B(\epsilon_S = 0.3)$ | 0.0230(0) | 0.0172(0) | **0.0169(0)** |
| | AUC | 0.8058(1) | 0.8507(0) | **0.8532(0)** |
| $t\bar{t}+h$ | $\epsilon_B(\epsilon_S = 0.7)$ | 0.2399(2) | 0.1794(1) | **0.1748(1)** |
| | $\epsilon_B(\epsilon_S = 0.3)$ | 0.0502(0) | 0.0357(0) | **0.0351(0)** |

Table 6: Results for the 4 top and top-top-Higgs signals: the areas under the ROC curve and the background efficiencies, at signal efficiencies of **70%** and **30%** respectively, correspond to the entire training dataset. Quoted uncertainties are extracted from three independent runs for each network architecture. Numbers in bold indicate the best performance.

|  |  | $\sigma$ | $\sigma_{\delta sys = 0.2}$ |
|---|---|---|---|
| PN | $\epsilon_S = 0.3$ | 32.18 | 7.63 |
| | $\epsilon_S = 0.7$ | 34.30 | 3.80 |
| PN$_{\text{int.}}$ | $\epsilon_S = 0.3$ | **37.53** | **10.27** |
| | $\epsilon_S = 0.7$ | 38.75 | 4.84 |
| PN$_{\text{int. SM}}$ | $\epsilon_S = 0.3$ | **37.86** | **10.44** |
| | $\epsilon_S = 0.7$ | **39.50** | **5.02** |
| ParT | $\epsilon_S = 0.3$ | 30.24 | 6.76 |
| | $\epsilon_S = 0.7$ | 33.48 | 3.62 |
| ParT$_{\text{int.}}$ | $\epsilon_S = 0.3$ | 36.82 | 9.90 |
| | $\epsilon_S = 0.7$ | 38.39 | 4.75 |
| ParT$_{\text{int. SM}}$ | $\epsilon_S = 0.3$ | **37.20** | **10.10** |
| | $\epsilon_S = 0.7$ | **39.05** | **4.91** |

Table 7: Significance table calculated for the top-top-Higgs signal.

# 6 Conclusions

In this work, we present a novel approach to event classification in particle physics by integrating physical information, in particular energy-dependent SM interactions, into advanced machine learning models. Our study focuses on improving transformer models with an attention matrix and graph networks such as PN with edge features, both of which reflect the dynamical nature of SM interactions.

The results show that PN and ParT exhibit superior performance when these pairwise features and interaction matrices are integrated. This integration improves background suppression by $10 - 40\%$ over the baseline models (PN without other physical information), with approximately **10%** of this improvement directly attributable to the SM interaction matrix. In a simplified statistical analysis, we find that these ML models increase significance by up to **30%** compared to the baseline model. To achieve a similar improvement in significance by increasing the luminosity $L$, one needs to increase $L$ by about **70%**, assuming that significance improvement scales with $\sqrt{L}$ when signal and background events are proportional to $L$. We conclude that embedding SM interactions as physical information in network structures is an important avenue in this field that could lead to more accurate and efficient event classification in particle physics.

# Acknowledgements

The author(s) gratefully acknowledges the computer resources at Artemisa, funded by the European Union ERDF and Comunitat Valenciana as well as the technical support provided by the Instituto de Física Corpuscular, IFIC (CSIC-UV). R. RdA is supported by PID2020-113644GB-I00 from the Spanish Ministerio de Ciencia e Innovación and by the PROMETEO/2022/69 from the Spanish GVA. RV is supported by the European Research Council (ERC) under the European Union's Horizon 2020 research and innovation programme (grant agreement No. 788223, PanScales).

# Appendix

## A   Additional Plots and Tables

This appendix complements the main text by providing an additional plot and two comprehensive tables. It summarizes the results for the entire 240k dataset, providing a complete perspective on the data's scope and the analysis outcomes.

### A.1   AUC

Fig. 5 displays the ROC curves for all architectures against various backgrounds, providing a visual perspective on their comparative performances throughout the entire dataset.

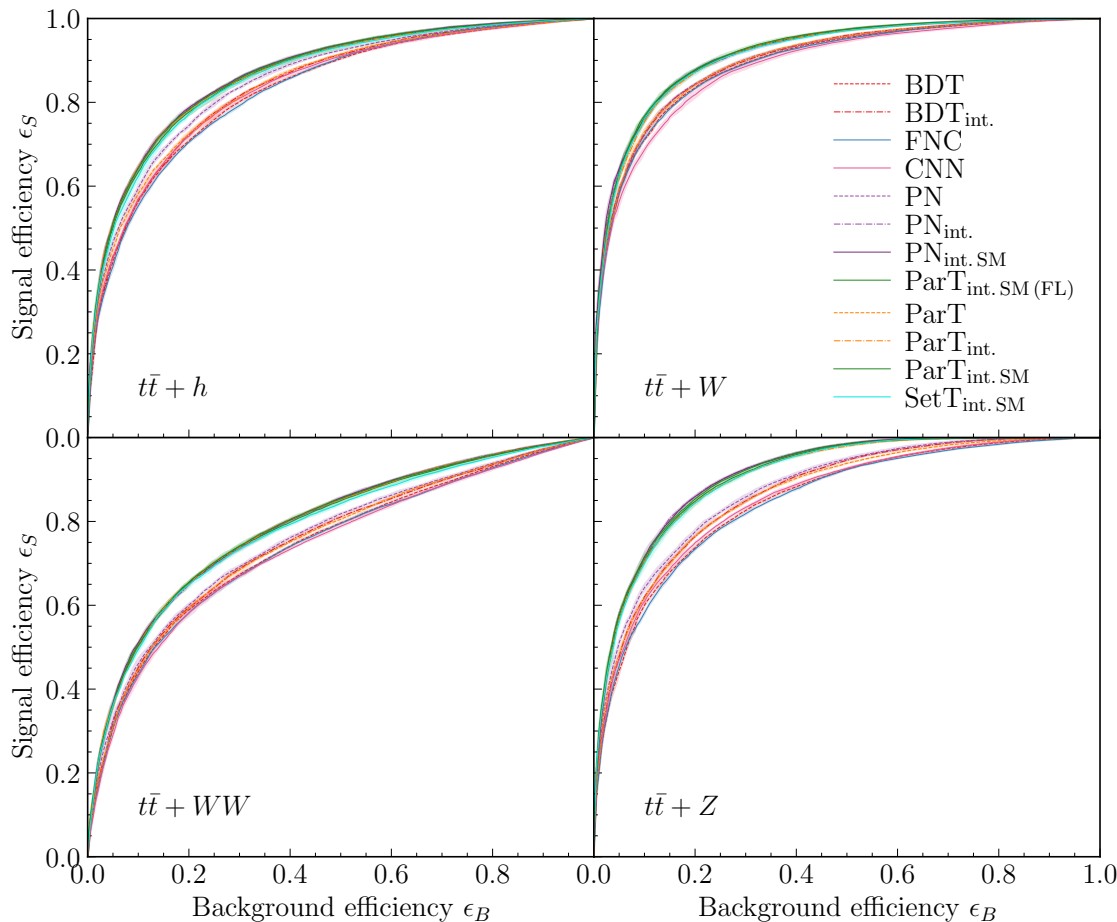

Figure 5: Receiver Operating Characteristic (ROC) curves for all architectures for the 4 top signal across the four background processes. The solid lines and error bands represent the mean and standard deviation of three independent runs for each architecture over the entire training dataset.

| | | BDT | BDT$_{int.}$ | FCN | CNN |
|---|---|---|---|---|---|
| $t\bar{t}+h$ | AUC | 0.833(0) | 0.840(0) | 0.832(0) | 0.838(3) |
| | $\epsilon_B(\epsilon_S=0.7)$ | 0.193(0) | 0.183(0) | 0.195(0) | 0.182(3) |
| | $\epsilon_B(\epsilon_S=0.3)$ | 0.022(0) | 0.022(0) | 0.023(1) | 0.021(2) |
| $t\bar{t}+W$ | AUC | 0.896(0) | 0.900(0) | 0.895(0) | 0.888(3) |
| | $\epsilon_B(\epsilon_S=0.7)$ | 0.093(0) | 0.087(0) | 0.093(1) | 0.107(3) |
| | $\epsilon_B(\epsilon_S=0.3)$ | 0.009(0) | 0.009(1) | 0.011(0) | 0.009(0) |
| $t\bar{t}+WW$ | AUC | 0.745(0) | 0.754(0) | 0.742(0) | 0.739(2) |
| | $\epsilon_B(\epsilon_S=0.7)$ | 0.339(0) | 0.317(2) | 0.341(0) | 0.344(2) |
| | $\epsilon_B(\epsilon_S=0.3)$ | 0.048(0) | 0.045(0) | 0.048(0) | 0.052(0) |
| $t\bar{t}+Z$ | AUC | 0.852(1) | 0.869(0) | 0.848(0) | 0.857(0) |
| | $\epsilon_B(\epsilon_S=0.7)$ | 0.167(1) | 0.149(2) | 0.170(2) | 0.161(2) |
| | $\epsilon_B(\epsilon_S=0.3)$ | 0.020(0) | 0.018(0) | 0.020(0) | 0.018(0) |

| | | PN | PN$_{int.}$ | PN$_{int.\ SM}$ | ParT$_{int.\ SM\ (FL)}$ |
|---|---|---|---|---|---|
| $t\bar{t}+h$ | AUC | 0.854(1) | **0.871(0)** | **0.872(0)** | 0.867(3) |
| | $\epsilon_B(\epsilon_S=0.7)$ | 0.161(2) | **0.129(1)** | **0.129(3)** | 0.138(4) |
| | $\epsilon_B(\epsilon_S=0.3)$ | 0.016(0) | 0.017(0) | 0.017(0) | **0.016(0)** |
| $t\bar{t}+W$ | AUC | 0.901(0) | 0.917(1) | **0.919(0)** | **0.918(1)** |
| | $\epsilon_B(\epsilon_S=0.7)$ | 0.089(1) | **0.072(1)** | **0.071(2)** | **0.071(2)** |
| | $\epsilon_B(\epsilon_S=0.3)$ | 0.008(0) | **0.007(0)** | **0.007(0)** | 0.007(0) |
| $t\bar{t}+WW$ | AUC | 0.759(2) | **0.791(0)** | **0.793(0)** | 0.791(3) |
| | $\epsilon_B(\epsilon_S=0.7)$ | 0.312(4) | 0.256(2) | 0.252(2) | **0.249(7)** |
| | $\epsilon_B(\epsilon_S=0.3)$ | 0.043(1) | 0.036(1) | 0.035(2) | 0.035(1) |
| $t\bar{t}+Z$ | AUC | 0.876(2) | **0.913(0)** | **0.913(0)** | 0.909(0) |
| | $\epsilon_B(\epsilon_S=0.7)$ | 0.139(5) | 0.095(1) | **0.094(2)** | 0.097(1) |
| | $\epsilon_B(\epsilon_S=0.3)$ | 0.015(0) | 0.013(0) | 0.012(0) | **0.010(0)** |

| | | ParT | ParT$_{int.}$ | ParT$_{int.\ SM}$ | SetT$_{int.\ SM}$ |
|---|---|---|---|---|---|
| $t\bar{t}+h$ | AUC | 0.843(1) | 0.869(0) | **0.871(0)** | 0.864(3) |
| | $\epsilon_B(\epsilon_S=0.7)$ | 0.179(3) | 0.131(2) | 0.132(0) | 0.141(4) |
| | $\epsilon_B(\epsilon_S=0.3)$ | 0.019(0) | **0.015(0)** | **0.015(0)** | **0.016(1)** |
| $t\bar{t}+W$ | AUC | 0.901(0) | 0.915(0) | **0.918(1)** | 0.915(2) |
| | $\epsilon_B(\epsilon_S=0.7)$ | 0.087(3) | 0.078(1) | **0.072(1)** | 0.074(2) |
| | $\epsilon_B(\epsilon_S=0.3)$ | 0.008(0) | 0.009(0) | 0.008(0) | 0.009(0) |
| $t\bar{t}+WW$ | AUC | 0.753(1) | **0.792(1)** | **0.792(1)** | 0.786(2) |
| | $\epsilon_B(\epsilon_S=0.7)$ | 0.318(5) | 0.250(2) | **0.248(2)** | 0.257(5) |
| | $\epsilon_B(\epsilon_S=0.3)$ | 0.047(1) | **0.032(0)** | 0.034(0) | 0.036(1) |
| $t\bar{t}+Z$ | AUC | 0.866(0) | 0.907(1) | **0.912(0)** | 0.907(2) |
| | $\epsilon_B(\epsilon_S=0.7)$ | 0.150(2) | 0.098(2) | **0.093(3)** | 0.100(4) |
| | $\epsilon_B(\epsilon_S=0.3)$ | 0.017(0) | 0.012(1) | **0.011(0)** | **0.011(0)** |

Table 8: The areas under the ROC curve and the background efficiencies, at signal efficiencies of **70%** and **30%** respectively, correspond to the entire training dataset (240k events). Quoted uncertainties are extracted from three independent runs for each network architecture. Numbers in bold indicate the best performance. In cases where the performances of multiple architectures are the best within the uncertainty, the results are both indicated.

## A.2   ttH

Comprehensive results covering other versions of the SM interaction matrices are presented in Table 9, which details the Area Under the Curve (AUC) for both the 4 top and top-top-Higgs signals. Specifically, the first row illustrates the AUC results for the 4 top signal, while the subsequent row delineates the outcomes related to the top-top-Higgs signal. It is important to note that, although the full dataset was employed for the 4 top signal analysis, the dataset used for the top-top-Higgs signal analysis was the same, yet it explicitly excluded data from the 4 top.

Five scenarios are provided for the PN and the ParT: the standard ParT/PN architecture, ParT/PN with the inclusion of pairwise features (int.), ParT/PN with the first iteration of the interaction matrix (SMids), ParT/PN with the second iteration of the interaction matrix (where the coupling constants on the SM are fixes parameters) and ParT/PN with the inclusion of SM running coupling constants (int. SM, which is the third iteration).

| | | PN | PN$_{\text{int.}}$ | PN$_{\text{int. SMids}}$ | PN$_{\text{int. SM const}}$ | PN$_{\text{int. SM}}$ |
|---|---|---|---|---|---|---|
| | AUC | 0.8471(1) | 0.8729(0) | 0.8725(0) | 0.8727(0) | **0.8739(0)** |
| $t\bar{t}t\bar{t}$ | $\epsilon_B(\epsilon_S = 0.7)$ | 0.1758(3) | 0.1387(1) | 0.1377(0) | 0.1384(0) | **0.1369(1)** |
| | $\epsilon_B(\epsilon_S = 0.3)$ | 0.0207(0) | 0.0182(0) | 0.0178(0) | 0.0178(0) | **0.0176(0)** |
| | AUC | 0.8146(2) | 0.8505(0) | 0.8489(1) | 0.8505(0) | **0.8523(0)** |
| $t\bar{t} + h$ | $\epsilon_B(\epsilon_S = 0.7)$ | 0.2292(1) | 0.1787(0) | 0.1785(1) | 0.1764(3) | **0.1733(1)** |
| | $\epsilon_B(\epsilon_S = 0.3)$ | 0.0471(1) | 0.0345(0) | 0.0343(1) | 0.0350(0) | **0.0340(0)** |
| | | ParT | ParT$_{\text{int.}}$ | ParT$_{\text{int. SMids}}$ | ParT$_{\text{int. SM const}}$ | ParT$_{\text{int. SM}}$ |
| | AUC | 0.8404(0) | 0.8708(0) | 0.8715(0) | 0.8717(0) | **0.8732(0)** |
| $t\bar{t}t\bar{t}$ | $\epsilon_B(\epsilon_S = 0.7)$ | 0.1842(3) | 0.1394(0) | 0.1389(2) | 0.1372(1) | **0.1366(0)** |
| | $\epsilon_B(\epsilon_S = 0.3)$ | 0.0230(0) | 0.0172(0) | 0.0180(0) | **0.0167(0)** | 0.0169(0) |
| | AUC | 0.8058(1) | 0.8507(0) | 0.8473(0) | 0.8497(0) | **0.8532(0)** |
| $t\bar{t} + h$ | $\epsilon_B(\epsilon_S = 0.7)$ | 0.2399(2) | 0.1794(1) | 0.1836(3) | 0.1801(1) | **0.1748(1)** |
| | $\epsilon_B(\epsilon_S = 0.3)$ | 0.0502(0) | 0.0357(0) | 0.0355(1) | 0.0367(0) | **0.0351(0)** |

Table 9: Results for the 4 top and top-top-Higgs signals: the areas under the ROC curve and the background efficiencies, at signal efficiencies of **70%** and **30%** respectively, correspond to the entire training dataset. Quoted uncertainties are extracted from three independent runs for each network architecture. Numbers in bold indicate the best performance.

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
