# Peer review of "Attention to the strengths of physical interactions: Transformer and graph-based event classification for particle physics experiments"

_SciPost Physics_

## Round 1 · Referee Report · Anonymous (Referee 1) · 2024-5-3

Report

The authors perform a comprehensive comparison of multiple event classification strategies in the context of 4 top searches (and later also for ttH measurements). They demonstrate how including hand-crafted information to the network architecture can increase performance and also introduce a new set of pairwise interactions inspired by SM interactions. The results are interesting and worth of a publication. However the additional SM-inspired interaction terms introduced by the authors does not seem to significantly improve the models compared to the versions that already include interactions such as the invariant mass and distances between particle pairs, as already explored in other publications. Indeed, compared to the models with these interactions, the improvements from the additional SM parameters are often below the 5% level for a number of metrics (unless I missed a key result), which questions their utility. In particular, what is the increase in computational complexity when these features are added (in terms of FLOPs, for example)? Would also be beneficial to study each set of newly introduced interaction terms independently, to identify which sets are providing useful additional information to the network architecture. The way the authors introduced the different interaction terms seem to indicate that there would be a comparison between each, but in all following plots only one version is shown. See more details below.

While not mandatory, the paper is a bit short on references to previous works presenting neural network architectures for tagging. I suggest using the living review (https://iml-wg.github.io/HEPML-LivingReview/) for suggestions of relevant literature to include.

L60: Maybe I’m missing the reasoning but why background samples such as tt+ jets are not included in this study? I would expect the contribution to 4tops (and later to ttH) to be important (such as ttbb for example).

L80: How do you define b-jets in this context? Based on reco-level tagging or based on gen objects matched to reco jets?

L136: BDTs compared to NNs are knows for their resilience against irrelevant features (also investigated in the context of anomaly detection in arXiv:2309.13111 and arXiv:2310.13057), so it seems surprising that a zero-padding strategy would result in degradation. Do you have an idea why that would be the case?

Additionally, both BDTs, MLPs, and CNNs would require some specific ordering of the inputs. How is that defined? Based on the pT of the objects? Sorry if I missed in the text.

L200: In the case of ParticleNet, what is the number of k neighbors used? In the space of jets and leptons, I imagine the number of constituents to be much smaller compared to previous studies using particles clustered to jets (which can go above 100 objects per jet). If the original number k=16 from ParticleNet is used, I imagine the graph is almost fully connected, is that correct?

Eq5: Have you tried including the embedding U as a concatenation rather than an addition? That would increase the complexity of the model, but could also lead to better use of these features

Eq6: The result matrix multiplication QK^{T} is a NxN matrix (N is the number of objects), however I would expect U to be of shape NxNxF with F being the number of pairwise features calculated for each of the pairs. How is that summed with the QK^{T} term?

L280: How is a 3-body invariant mass calculated from a pair ij?

Sec. 4.2: Some of the interaction terms introduced require the knowledge of a jet being a b or a light jet. If there is a mistag and a b-jet is wrongly identified (or a light jet is misidentified), does that affect the performance?

L283: Isn’t \delta R_{ij} already somewhat included in the edge information of particlenet in Eq.5? the x_i - x_j term, besides the norm, should also carry the same information. Did the inclusion of \delta R_{ij} bring any benefit to the particlenet implementation?

L338: The large negative number is understood in the PartT implementation using attention, but is that also used for the ParticleNet application? There there was no softmax operation applied, unless that was introduced as part of the interaction matrix formulation in this work?

Fig1: This figure is really interesting and I partially agree with the authors conclusions. The AUC is often too simplistic to provide the complete picture and even if the AUC does not improve dramatically, the signal efficiency for fixed background efficiency can still change by a good margin. Would be great to have a similar plot but instead of the AUC using the signal efficiency at fixed background efficiency (say 30% or 70%), which is also closer to how these classifiers would be used in a realistic analysis.

Fig2: The background rejection improvement with respect to a baseline model is a great distribution to show, however I do not understand why the 48k data sample is chosen to be the baseline comparison. Is there some specific restriction in a real analysis that would only allow around 50k simulated samples to be available for training? Given the focus of the paper on the improvements obtained by adding SM inspired quantities, I would use as a baseline the particlenet results with full data but without any interaction terms. That would also help the reader navigate the results to see the improvements brought by the different choices of interaction terms added.

L383: Similar to the previous comment, the numbers described as improvements wrt the baseline do not seem very meaningful unless there is a strong reason why a dataset with only 50k samples is needed as the baseline.

Tab: 5: I appreciate the comparison of the predicted significances and how they are modified in the presence of a systematic uncertainty affecting the background processes. In this context, using the signal efficiency value that maximizes the SIC curve (sig. eff/ sqrt(bkg. eff) vs sig. eff.) would perhaps be a better/more realistic choice. How the expected significance changes if instead of fixing the signal efficiency one used the maximum SIC point? That should also give you the best expected significance for each classifier, which is always good to know.

Fig.4 Again, these results would be more interesting when evaluated over the full dataset instead of only part of the data.

Recommendation

Ask for major revision

  • validity: good
  • significance: good
  • originality: good
  • clarity: high
  • formatting: excellent
  • grammar: excellent

Author:  Polina Moskvitina  on 2025-01-07  [id 5091]

(in reply to Report 1 on 2024-05-03)
Category:
answer to question

Dear Referee,

We would like to thank you for your detailed feedback on our manuscript. We have attached a PDF file containing a point-by-point response to your comments.

Attachment:

ReplyToReferee3.pdf

---

## Round 1 · Referee Report · Anonymous (Referee 2) · 2024-5-22

Strengths

1 Novel method to improve HEP event-level classifiers based in physics information
2 Public dataset provided
3 Thorough comparison of the novel method in several state-of-the-art algorithms
4 Application to relevant examples

Weaknesses

1 Clarity can be improved in several places (see requested changes)
2 Improvement from novel method on top of existing strategies of the order of 10%

Report

The paper presents a novel way to use physical information to improve event-level classifiers for high-energy-physics searches and/or measurements of rare processes and applies them to the relevant example cases of 4-top and tth production at the LHC. As such, it provides a new pathway in this key aspect of HEP data analysis with a clear potential for follow-up work. In particular, the proposed mechanism may be used in data analysis in HEP experiments, such as the experiments at the LHC and may be especially useful in the usual case of limited training statistics of signal and background samples in the relevant pre-selected phase space. The paper is in general well written, provides sufficient detail, citations and a clear summary, abstract and introduction. Besides a small number of comments to improve the clarity of the description, my main criticism is that abstract, introduction and summary can benefit from a clearer description of the presented work in the broader scientific context. While the proposed mechanism to inform the classifier about the existence and strength of Standard Model interaction vertices is new, interesting and - as shown in the manuscript - helpful for the training of the classifier, the idea of including physics information based on two-particle invariant masses and angular distance is not new and this can be more clearly described to strengthen the document (see the requested changes below). After my comments have been satisfactorily addressed, I recommend publication in SciPost Physics.

Requested changes

Major:

1) I suggest to improve the clarity concerning the use of physics information in HEP classifiers in the abstract, introduction and conclusions to better highlight the novelty of the proposed with respect to well-established and other studied approaches. Here are a few concrete suggestions but I also invite the authors to read through the relevant paragraphs again and improve them where they see fit:

  • lines 43-45: “These methods are mainly used in the context of the classification of jet data. On the other hand, their application to event-level data has not yet been explored to the same degree, and BDTs are still the most commonly used method.” - It is true that there are analyses that use BDTs as event level classifier, but there are numerous examples where deep learning architectures are used for this purpose. I suggest to rephrase this sentence and provide references to examples of HEP publications that use neural networks. In this context, I also suggest to discuss in the introduction (and in Section 3.1) that the use of high-level features is a standard technique in BDTs in HEP. In the same spirit for line 269: As it is common practice to use designed features as the input to BDTs, please provide citations to example publications of HEP Collaborations, such as ATLAS, CMS, ALICE and/or LHCb, instead of or in addition to Ref. [32].

  • lines 46-53: The use of physical information in the context of deep learning has been discussed in several papers before. Actually, such papers are cited in lines 268-272. In my opinion, the discussion of this previous work should be moved to a central part in the introduction of the paper to better lay the ground for the presented work.

  • lines 465-468: I suggest that the authors consider rephrasing the last sentence of the conclusions to highlight the potential gain of their novel SM interaction matrices as opposed to a general statement about the inclusion of physics information in networks, which had already been discussed in previous literature (such as Refs. [33-36]).

2) line 138: Isn’t it a strong limitation to remove information about possible additional same-charge, same-flavor leptons from the BDT training, in particular for same-sign, same-flavor, 3-lepton and 4-lepton events? Same comment for the FCN (lines 158-159).

3) Table 5: I am surprised that the focal loss ParticleTransformer performs very well in terms of significance. How is this related to the comparatively bad discrimination of this classifier in Figure 3, where it seems to separate the background and the signal much worse than for example the BDTs?

4) lines 418-420: I do not understand this part of the discussion, as the significances in the case of 20% systematic uncertainty on the background are compared, but the corresponding increase in statistics seems to be calculated neglecting systematic uncertainties, if I am not mistaken. Please clarify. Same comment for lines 447-449 for tth and lines 463-464 in the conclusions.

5) Conclusions: “10% of the improvement directly attributable to the SM interaction matrix.” - If I compare all background rejections of “int.” and “int. SM” in Table 6, I do not see in any of the numbers an improvement near 10%. The same is true for the background rejections in Table 4. Or do you mean that it is 10% of the overall 10-40% improvement, i.e. only 1-4%? If this is the case, please rephrase the conclusions and the abstract for more clarity.

Minor (typos/clarifications/etc.):

1) lines 113-114: “The dataset includes 302 072 events, half of which correspond to the four tops signal and half of which are background processes.” - If I multiply the numbers from Table 1, I get for 4 tops: 32,463,742 * 0.007 = 230k, which is much more than half of the 300k events in the dataset. Please clarify.

2) line 115: “All background processes have an equal number of events.” - Please clarify this statement, as it seems to be not consistent with the numbers in Table 1.

3) line 153: I suggest to cite the Dropout paper.

4) line 154: I suggest to provide the values of the default parameters.

5) line 161: In my view, Ref. [21] is not appropriate as a general reference for the idea of convolutional neural networks. I suggest to cite original work on CNNs here.

6) line 169: Please clarify why the maximum number of particles per event is 18, as each top decays to three particles, resulting in a maximum of 12 particles.

7) Table 2: Please explain the meaning of “γ_tag”, as it was not introduced before.

8) line 184: I would expected that also the particle type is part of the node features. Please clarify.

9) line 192: Please provide more details about the “attention-weighted procedure”. How do you define the attention heads?

10) line 247: Please check the grammar in “i.e. pt were is large” at the end of the bullet.

11) line 327: Do you reduce nf to values smaller than 6 if the scale is below the top quark mass? Please clarify.

12) Figure 1: I suggest to add the focal loss ParticleTransformer and the SetTransformer curves to this figure, as you include these in other relevant parts of the results section, such as Table 4, Figure 3 and Table 5.

13) Figure 1: Typo in “FNC” -> “FCN”. I also suggest to remove the label “no pair int.” from the legend for consistency with Table 3.

14) Figure 2: It is curious that the “int. SM” models clearly outperform the benchmark ParticleNet for tth, ttWW and ttZ but for ttW the improved is not that pronounced, in particular not for lower signal efficiencies. It would be instructive to discuss the origin of this behavior in the body of the text. Is this for example connected to the share of ttW in the total background or to the similarity of ttW to the 4 top signal?

15) Figure 3: What is the “total cross section”? Is it sum the of the SM signal and background cross sections? Please clarify.

16) Figure 3: Typo in “FNC” -> “FCN”.

17) line 431: Should “ongoing couplings constants” rather be “running coupling constants”?

18) I suggest to reference Table 8 somewhere in text of the Appendix.

Recommendation

Ask for minor revision

  • validity: top
  • significance: good
  • originality: high
  • clarity: high
  • formatting: perfect
  • grammar: perfect

Author:  Polina Moskvitina  on 2025-01-07  [id 5090]

(in reply to Report 2 on 2024-05-22)
Category:
answer to question

Dear Referee,

We would like to thank you for your detailed feedback on our manuscript. We have attached a PDF file containing a point-by-point response to your comments.

Attachment:

ReplyToReferee2.pdf

---

## Round 1 · Referee Report · Tilman Plehn (Referee 3) · 2024-6-13

Report

The paper should definitely be published, in that it asks a highly interesting and relevant question. It is also well written, and the documentation of the results is very carefully done. I have only very few comments (ordered by appearance in the paper): 1. Please make sure that relevant works on ML-tagging at the LHC are cited. References is what gives our young people jobs; 2. Please explain the attention layer a little more in detail, especially in view of the set transformer mentioned later. We should not assume that all readers know, for instance, the differences between a graph network and a transformer; 3. Eq.(4) is a very sloppy version of a formula. It would be nice to first write the BCE loss and then the focal loss, including a definition of all symbols in a way that a student can just implement it; 4. In Sec.3 I am missing a discussion of network size and training data efficiency. I would expect the methods to be very different here, will come back to this for Fig.1; 5. In 4.1 the natural question arises - what is physics information and what is just a covariant representation under a known symmetry. Please comment and separate this carefully; 6. In Fig.1, please find a way to label the curves such that the reader does not have to spend significant time going back and forth between curves and labels. Line styles combined with colors might work. 7. Content-wise, I have a hard time understanding Fig.1. Why are PN and ParT without physics information so bad, comparable to the BDT? Or is the BDT good because the problem is simple. And if that is the case, why does the physics information help? I am confused, some of the results are really counter-intuitive... 8. Also in Fig.1, is the increase with more training data really the same for the graph and the transformer? Is that not against our general expectations, and also against the experience with the pretrained ParT? 9. I do not understand the comment at the very end of 5.2, that the PN is similar to the ParT; 10. Finally, the obvious question is if this improvement can be translated to the high-performance results for jet tagging. And since the highest-performing taggers to date are covariant, what do the authors expect to happen for those? As you can see, I really enjoyed reading the paper, but maybe the authors can think about some of my questions, assuming that other readers might end up having similar questions. And then, SciPost should definitely publish this very nice study!

Recommendation

Ask for minor revision

  • validity: -
  • significance: -
  • originality: -
  • clarity: -
  • formatting: -
  • grammar: -

Author:  Polina Moskvitina  on 2025-01-07  [id 5089]

(in reply to Report 3 by Tilman Plehn on 2024-06-13)
Category:
answer to question

Dear Tilman Plehn,

We would like to thank you for your detailed feedback on our manuscript. We have attached a PDF file containing a point-by-point response to your comments.

Attachment:

ReplyToReferee1.pdf

---

## Editorial Decision

resubmitted